# EFFICIENT FINE-TUNING WITH DECOMPOSED FOUNDATION MODEL

## ABSTRACT

Fine-tuning billion-scale large language models (LLMs) is challenging due to the extremely large model size, particularly in memory-constrained scenarios, even with parameter-efficient fine-tuning (PEFT) and quantization. To address this challenge, we propose a novel method based on the *decomposition then fine-tuning* (DeFT) paradigm, which effectively decomposes the foundation model and reduces the number of model parameters during fine-tuning, while retaining model quality. DeFT introduces a highly efficient layer importance aware search algorithm for fine-grained model decomposition and successfully repurposes model decomposition for fine-tuning. Additionally, DeFT can seamlessly integrate with PEFT and quantization methods to enhance fine-tuning efficiency further. Extensive experiments on various LLM backbones demonstrate that DeFT achieves comparable or even better performance than the baseline PEFT and quantization methods, while improving both memory efficiency and computation efficiency for fine-tuning. Remarkably, DeFT enables fine-tuning of a 65B model on a consumer GPU with just 24GB of memory, all without relying on offloading strategies, saving significant expenses for purchasing or renting high-end GPUs.

## 1 INTRODUCTION

Transformer-based language models have been extensively studied since the proposal of the self-attention mechanism (Vaswani et al., 2017) and the foundation of the pre-training paradigm (Peters et al., 2018; Devlin et al., 2019). Following the scaling law (Kaplan et al., 2020), modern large language models (LLMs) have billions of model parameters for better predictive accuracy (Brown et al., 2020; Zhang et al., 2022; Touvron et al., 2023). Consequently, full fine-tuning of such large LLMs is extremely expensive due to the required computing resources and time consumption.

To reduce the cost of LLM fine-tuning, researchers have proposed parameter-efficient fine-tuning (PEFT) techniques, where a large portion of the model parameter is frozen and only a very small part of the parameters needs to be updated (Houlsby et al., 2019; Hu et al., 2022; Li & Liang, 2021; Zaken et al., 2022). These works achieve competitive predictive accuracy while reducing memory costs compared with full fine-tuning. To further cut down memory footprint, researchers propose to incorporate quantization into fine-tuning by storing the foundation model in low-bit floating point numbers (*e.g.,* 4-bit). QLoRA (Dettmers et al., 2023) and LoftQ (Li et al., 2024) are two representative approaches. Despite the success of PEFT and quantization-aware fine-tuning in reducing memory consumption, the foundation model sizes remain

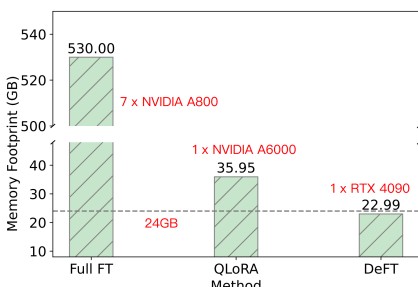

Figure 1: Memory cost of fine-tuning a 65B model of different methods.

unchanged, and exorbitant memory consumption by tens of billions of model parameters poses considerable challenges to fine-tuning, especially in memory-constraint scenarios (Liao et al., 2023).

In this work, we propose a novel method based on the *decomposition then fine-tuning* paradigm (namely DeFT), which can be flexibly integrated with PEFT and quantization to further improve the memory efficiency and computation efficiency. DeFT first conducts model decomposition to reduce

the number of foundation model parameters and then fine-tunes the decomposed model to accommodate the downstream tasks. To mitigate the decomposition overhead and boost the model quality, DeFT exploits (i) activation-aware singular value decomposition (SVD), by taking advantages of the closed-form property of compression loss in SVD (Eckart & Young, 1936; Wang et al., 2024) to provide a fast evaluation on the reconstruction error; (ii) a highly efficient search algorithm to enable fine-grained decomposition, built on top of our detailed analysis of model decomposition. Furthermore, we optimize DeFT to facilitate usabilities, such as cache mechanisms and selective model loading/quantization. Therefore, DeFT enables fine-tuning a 65B model on a consumer GPU with 24GB of memory, as shown in Figure 1, demonstrating its practical value in memory-constrained scenarios. Additionally, with DeFT, we can save significant expenses for purchasing or renting high-end GPUs. To summarize, our main contributions in this paper are as follows.

- We propose a novel fine-tuning method based on the *decomposition then fine-tuning* paradigm (DeFT). It first introduces a highly efficient layerwise importance aware search algorithm for fine-grained foundation model decomposition, and then fine-tunes the decomposed model.

- DeFT is seamlessly incorporated with representative PEFT and quantitation-aware fine-tuning methods, and extensive experiments are carried out to demonstrate repurposing model decomposition for fine-tuning. DeFT effectively reduces the number of foundation model parameters while achieving comparable or even better performance than the baselines.

- DeFT showcases the memory efficiency and computation efficiency benefits for fine-tuning. Notably, it enables fine-tuning a 65B model on a consumer GPU with 24GB memory without using offloading, saving significant costs associated with buying or renting high-end GPUs.

## 2 METHODOLOGY

The success of quantization-aware fine-tuning inspires us to explore other model compression techniques to further improve fine-tuning efficiency. To make the compression effective for fine-tuning, two major concerns must be addressed: (i) the overhead of model compression must be small enough to achieve efficiency gains in terms of the end-to-end fine-tuning time cost; (ii) fine-tuning performance degradation needs to be limited to an acceptable range.

To address these two concerns, model decomposition could be an appropriate solution. It is a matrix decomposition technique that can be executed in a one-step process, significantly reducing excessive overhead. Moreover, its mathematical guarantee makes it easy to estimate fine-tuning performance through theoretical compression loss. Singular value decomposition (SVD) has been extensively studied and proven to be a practicable solution for model decomposition (Saha et al., 2023; Yuan et al., 2023; Wang et al., 2024). However, its potential to in fine-tuning remains unexplored. To this end, we propose a novel method, DeFT, that can effectively incorporate model decomposition to reach a graceful balance between fine-tuning performance and efficiency. In this section, we first introduce the workflow of DeFT. After that, we discuss the feasibility of repurposing model decomposition for fine-tuning and elaborate on the technical details of DeFT.

### 2.1 NOTATIONS AND THE WORKFLOW OF DEFT

The overview of DeFT is shown in Figure 2, and its workflow is particularized as follows: (i) DeFT constructs calibration data from the downstream task dataset. (ii) DeFT collects the input feature $X$ and outlier weighted layer importance for each layer $a^l$ and then obtains the Cholesky decomposition of $XX^T$, denoted as $S$. Subsequently, it decomposes $WS$ with SVD, and the inverse of scaling matrix, *i.e.,* $S^{-1}$, are absorbed into the $V$ matrix, where $W \in \mathbb{R}^{d \times d}$ is a pretrained weight, $d$ is the hidden size of the pretrained model, and $W \simeq W' = U\Sigma V S^{-1}$. (iii) For an LLM with $n$ layers, DeFT searches for the best truncation positions $\theta_l$ for each layer $l$ with an efficient layer importance aware algorithm (See equation 3 - equation 6 for more details), and the results are cached on disks. (iv) DeFT loads the truncated singular values, leverages their tail parts, *i.e.,* $A \in \mathbb{R}^{r \times d}$ and $B \in \mathbb{R}^{d \times r}$, to initialize the LoRA module, and utilizes the rest, *i.e.,* $W_u \in \mathbb{R}^{d \times (\theta_l - r)}$ and $W_v \in \mathbb{R}^{(\theta_l - r) \times d}$, to replace the pre-trained weights. (v) Fine-tuning starts. The decomposed foundation model is frozen, and only the LoRA module $A^*$ and $B^*$ is trainable. (vi) After fine-tuning, the weight difference, *i.e.,,* the adapter weight compared to the original pre-trained weight, can be obtained by $W_u W_v + B^* A^* - W$.

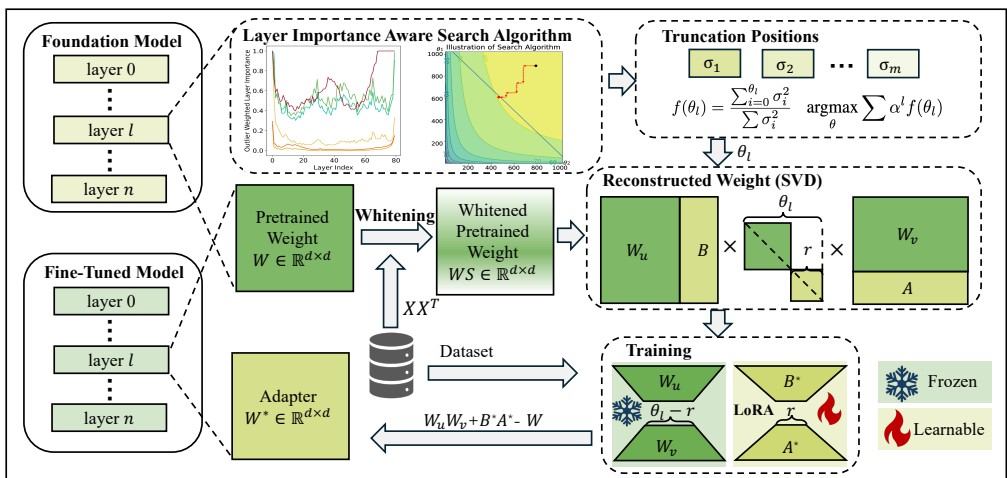

Figure 2: The overview of DeFT.

## 2.2 REPURPOSE MODEL DECOMPOSITION FOR FINE-TUNING

Conventional truncated SVD provides mathematical proof for the closed-form solution of the compression loss, *i.e.,* Eckart-Young Theorem (Eckart & Young, 1936), which is a reliable method to directly measure the reconstruction error of a matrix and its low-rank approximation:

$$L = \|W - W'\|_F, \tag{1}$$

where $W$ is a weight matrix, and $W'$ is the low-rank approximation of $W$. When it comes to model compression, although the vanilla truncated SVD can accomplish the model decomposition, it suffers from significant performance degradation since it does not consider the distribution of inputs and outputs (Yu & Wu, 2023; Yuan et al., 2023).

Recent research regarding SVD in LLM decomposition leverages activation to mitigate reconstruction error brought by outliers (Yuan et al., 2023; Yu & Wu, 2023). Wang et al. (2024) propose a whitening technique to capture data distribution of inputs. It first collects inputs $X$ and then obtains the Cholesky decomposition of $XX^T$, denoted as $S$. Subsequently, $WS$ is being decomposed with SVD, where the compression loss $L$ is formulated as the following equation 2 instead of equation 1.

$$L = \|WX - W'X\|_F \tag{2}$$

Moreover, it gives the mathematical proof for the closed-form of the compression loss, *i.e.,* Theorem 1, offering an efficient and reliable way to assess model quality that requires only theoretical calculations of the compression loss $L$, rather than expensive benchmarking.

**Theorem 1.** *(Wang et al., 2024) Given an input $X$ and a weight matrix $W$, let $S$ be the Cholesky decomposition of $XX^T$ and its singular value decomposition results $U\Sigma V^T$ derived from applying SVD to $WS$. The activation-aware compression loss of truncating the smallest singular values $\{\sigma_{m+1}, \sigma_{m+2}, ...\sigma_k\}$ is $L^2 = \|WX - W'X\|_F^2 = \|\sum_{i=m+1}^{k} \sigma_i u_i v_i^T S^{-1} X\|_F^2 = \sum_{i=m+1}^{k} (\sigma_i)^2$ and such truncating leads to the lowest loss, where $u_i$ and $v_i$ are the $i$-th left singular value and right singular value respectively.*

Many existing approaches uniformly compress all the layers under a preset compression rate, overlooking the varying compression sensitivity of different layers (Wang et al., 2024). However, sensitivity differences exist among layers (Geva et al., 2021; Sharma et al., 2023). This inevitably introduces unnecessary reconstruction errors, which could be extremely fatal for fine-tuning. The model reconstruction error could be too large to make fine-tuning converge, preventing it from achieving performance comparable to that of conventional fine-tuning methods.

To repurpose model decomposition for fine-tuning, a fine-grained search for layerwise truncation positions is essential, as models with lower reconstruction errors tend to yield higher accuracy on downstream tasks. To this end, we propose a *decomposition then fine-tuning* (DeFT) method, which models layer importance with layerwise outliers distribution and exploits it for fine-grained foundation model decomposition, reaching a graceful balance between performance and efficiency.

## 2.3 FORMULATION OF FINE-GRAINED DECOMPOSITION

According to Theorem 1, we can leverage singular values of a certain matrix to compute its corresponding reconstruction error under specific truncation positions. Thus we can define the performance score of layer $l$ through:

$$f(\theta_l) = \frac{\sum_{i=0}^{\theta_l} \sigma_i^2}{\sum \sigma_i^2},$$ (3)

where $\theta_l \in \mathbb{Z}^+$ denotes the SVD truncation position for layer $l$, and $\sigma$ is the singular values of layer $l$. The larger the performance score is, the less reconstruction error is.

To enable fine-grained truncation position configurations, we introduce layerwise outlier distribution (Yin et al., 2024) as a coefficient to balance the memory budget allocation among layers that have different sensitivity to compression. It is proven to be effective in modeling layer sensitivity (Yin et al., 2024) by computing the ratio of outliers in the activations (output features) of an LLM:

$$\alpha^l = \frac{\sum_{i=1}^{N} \sum_{i=1}^{M} \mathbb{I}(\mathbf{A}_{ij}^l > T\bar{\mathbf{A}}^l)}{M \times N},$$ (4)

where $\alpha^l$ represents the outlier weighted importance for layer $l$; $N$ and $M$ represent the input and output channel of the pre-trained weight matrix, respectively; $\mathbf{A}^l$ is the absolute values of activation outputs of layer $l$; $\bar{\mathbf{A}}^l$ is the mean of $\mathbf{A}^l$; $\mathbb{I}(\cdot)$ denotes an indicator function returning 1 if $\mathbf{A}_{ij}^l$ is larger than $\bar{\mathbf{A}}^l$ else 0; and $T$ is a hyperparameter which is set to 5 following (Yin et al., 2024). Then, the truncation position selection problem can be formalized as follows.

### 2.3.1 PROBLEM DEFINITION

For a large language model, given its layers $l \in \mathbf{L}$, layers' corresponding performance scorer function $f$, memory consumption function $g$, and layer importance $\alpha^l$, to fit the compressed model into a limited memory $\mathcal{B}$, the truncation position selection algorithm finds truncation positions for each layer, *i.e.,* $\theta_l$, where it has the maximal weighted sum $\alpha^l f(\theta_l)$ while satisfying the memory constraint and performance function lower-bound constrain:

$$\underset{\theta}{\text{argmax}} \sum \alpha^l f(\theta_l)$$
$$s.t. \sum g(\theta_l) \leq \mathcal{B}$$ (5)
$$f(\theta_l) \geq \mathcal{P}_l,$$

where $\mathcal{P}_l$ is the lower bound of performance score at layer $l$. This optimization problem is a typical integer programming problem with a vast solution space, where the exhaustive search is infeasible. Therefore, we propose an approximate algorithm to get a solution that achieves good performance and is efficient.

Figure 3: (a) A surface spanned by solutions, *i.e.,* Cartesian product of $\theta_1$ and $\theta_2$, where surface represent values of $f(\theta_1) + f(\theta_2)$. (b) Performance score for each "v_proj" layer (the hidden size is 8192) from LLaMA-65B, which is computed by Equation equation 3 and normalized into [0, 1]. The Y-axis "Layer position" denotes the $i$-th transformer block.

## 2.4 SOLUTION SPACE OF TRUNCATION POSITIONS SELECTION

According to equation 3, the objective function is a weighted sum of the performance scores, whose values are located on a hypersurface in a high dimensional space and the constraints define a set of boundaries. To better understand this concept, we visualize the solution space in Figure 3(a) with a simplified objective function including two performance score functions.

We visualize performance scores defined by equation 3 in Figure 3(b). For singular values of a specific layer, the square sum of its head part occupies the largest proportion of the total and it showcases the marginal increment when accumulating the tail part, exhibiting a strong long-tail

distribution. This indicates layers can be compressed to a large extent with only small compression loss. Moreover, from the overview of the performance scores, we can observe the differences among layers. Such differences strongly suggest that truncation positions should use layerwise selection rather than a uniform setting.

Here is another intuitive observation that can help identify where the optimal solution should be located: the more parameters are preserved, the less reconstruction error of the model is, and too many preserved parameters could lead to dissatisfaction with the memory budget constraint. Therefore, the inequation constraint regarding memory budget can be rewritten into an equation constraint. Since our problem is an integer programming problem, the optimal solution should be located on or very close to the hyperplane. Based on this, we design an approximate algorithm that starts outside the feasible domain and stops once entering the feasible domain, *i.e.,* crossing the hyperplane.

## 2.5 SEARCH FOR THE MOST PROFITABLE TRUNCATION

Initially, we force performance scores of each layer $l \in \mathbf{L}$ equal to a very high value, *i.e.,* 0.999. Generally, this leads to dissatisfaction with the constraint $\sum g(\theta_l) < \mathcal{B}$. Then, the algorithm works in an iterative manner. In each iteration, the algorithm evaluates the effect of truncation position reduction of each layer:

$$
\begin{aligned}
\text{loss}_l &= (1 + \eta \alpha_l) \frac{\triangle f(\theta_l)}{\triangle g(\theta_l)} \\
&= (1 + \eta \alpha_l) \frac{f(\theta_l) - f(\theta_l - G)}{g(\theta_l) - g(\theta_l - G)},
\end{aligned} \tag{6}
$$

where $\eta$ is a coefficient to scale the impact of layer importance, and $G$ is the granularity that limits the feasible truncation positions.

Figure 4(a) illustrates the mechanism of how this metric works. It assesses the collaborative effects of variations in the performance scores and memory budget consumption. Moreover, it incorporates outlier weighted layer importance as a coefficient to penalize compression over

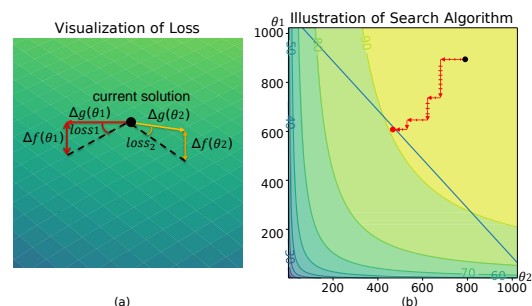

(a)                        (b)

Figure 4: (a) Visualization of the loss. (b) Demo of how the search algorithm works, where the blue line denotes the hyperplane derived by the budget constraint, and the algorithm starts at the black dot and stops at the red dot. These two figures illustrate a simplified scenario, where only two different dimensions are under consideration and the outlier weighted layer importance is ignored.

sensitive layers. The larger $\text{loss}_l$ is, the more likely the truncation position reduction is to damage the overall performance. In each step, the algorithm selects the move that leads to the minimal negative effects and applies it. The algorithm stops once it reaches or goes across the hyperplane, where layers that only have trivial parameter reduction remain the same. A simplified algorithm demo is presented in Figure 4(b), and Algorithm 1 in Appendix A outlines the search process.

Note that through this approximate algorithm, we can perform the search process within a few seconds. Besides, the model decomposition can also be finished in a short time. For example, for a 7B model, it takes about 10 minutes for model decomposition, and the decomposition results can be cached on the local disk and reused later.

## 2.6 MECHANISMS TO FACILITATE USABILITY

In addition to leveraging model decomposition to improve efficiency, we further optimize DeFT to improve its usability on devices that have limited memory resources.

**Cache Mechanism** We design a cache mechanism to reduce the overhead of DeFT, where singular vectors are decomposed offline and cached on the disks for reuse. That means, for different settings of compression rates, we do not need to repeat the SVD decomposition process, saving notable computation cost. The decomposition results are used as input for the search algorithm to find the best truncation positions. For each start of fine-tuning, DeFT first reads user-defined constraints and performs the search algorithm, which can be finished within a few seconds or a few minutes. Once

Table 1: Fine-tuning performance on the arithmetic reasoning tasks. Full FT: full fine-tuning.

| Models | Methods | #Params | #Trainable | AddSub | SingleEq | MultiArith | SVAMP | GSM8k | Avg. |
|---|---|---|---|---|---|---|---|---|---|
| GPT-3.5$_{175B}$ | - | - | - | 56.40 | 69.90 | 83.80 | 88.10 | 85.30 | 76.70 |
| LLaMA-7B | Full FT | 6.74B | 6.74B | 82.04 | 76.97 | 79.83 | 48.40 | 32.22 | **63.89** |
| | QLoRA | 6.74B | 72M | 79.66 | 80.84 | 78.01 | 44.93 | 30.2 | 62.73 |
| | +DeFT | 5.73B | 72M | 82.28 | 80.01 | 76.19 | 46.63 | 29.39 | 62.90 |
| LLaMA-65B | Zero shot | 65.29B | 0M | 2.03 | 0.98 | 1.26 | 1.90 | 2.20 | 1.67 |
| | QLoRA | 65.29B | 357M | 85.49 | 91.20 | 86.41 | 71.40 | 59.04 | 78.71 |
| | +DeFT | 55.49B | 357M | 90.46 | 92.39 | 86.98 | 75.67 | 58.70 | **80.84** |
| LLaMA-2 13B | Zero shot | 13.02B | 0M | 11.14 | 16.14 | 9.24 | 12.00 | 7.20 | 11.14 |
| | Full FT | 13.02B | 13.02B | 88.61 | 91.34 | 87.39 | 68.60 | 53.75 | **77.94** |
| | QLoRA | 13.02B | 112M | 83.04 | 89.17 | 84.87 | 63.00 | 48.29 | 73.67 |
| | +DeFT | 11.06B | 112M | 84.3 | 89.57 | 83.19 | 66.20 | 45.11 | 73.67 |
| | LoRA | 13.02B | 112M | 86.58 | 91.34 | 84.87 | 69.00 | 52.77 | 76.91 |
| | +DeFT | 11.06B | 112M | 87.09 | 92.52 | 86.55 | 67.30 | 47.31 | 76.15 |
| LLaMA-3 70B | QLoRA | 70.55B | 372M | 93.67 | 94.29 | 92.44 | 85.30 | 74.91 | **88.12** |
| | +DeFT | 60.10B | 372M | 92.15 | 95.28 | 91.18 | 84.50 | 75.74 | 87.77 |
| Qwen-2.5 7B | Full FT | 7.62B | 7.62B | 91.65 | 93.11 | 92.02 | 88.40 | 78.54 | **88.74** |
| | QLoRA | 7.62B | 74M | 91.90 | 95.87 | 91.60 | 86.00 | 72.63 | 87.60 |
| | +DeFT | 6.51B | 74M | 93.16 | 95.08 | 90.34 | 83.40 | 70.13 | 86.42 |
| | LoRA | 7.62B | 74M | 93.42 | 95.08 | 92.86 | 84.70 | 73.46 | 87.90 |
| | +DeFT | 6.51B | 74M | 93.16 | 96.06 | 93.28 | 84.70 | 71.42 | 87.72 |
| Qwen-3 32B | QLoRA | 32.76B | 241M | 92.66 | 95.67 | 93.70 | 86.60 | 78.54 | **89.43** |
| | +DeFT | 28.00B | 241M | 92.15 | 95.28 | 94.96 | 85.10 | 79.45 | 89.39 |

the algorithm stops, it dynamically selects the desired singular vectors of each layer according to the search results, and caches them on the disks.

**Selective Model Loading/Quantization**   Existing practice replaces the original weight with its decomposed one. For LLMs that have over tens of billions of model parameters (*e.g.,* LLaMA-65B), it is impossible to load the whole model into a single device with limited memory even under 4-bit quantization (Dettmers et al., 2023). To bridge this gap, we optimize model loading, preventing the original pre-trained weight from being loaded or quantized in advance, but straightforwardly loading and quantizing its corresponding decomposed singular values. With its help, we can successfully fine-tune a 65B model on a consumer GPU with 24GB of memory.

## 3 EXPERIMENTS

### 3.1 EXPERIMENT SETUPS

**Foundation Models and Baselines**   Foundation models used in our experiments include the LLaMA family (Touvron et al., 2023), Mistral-7B v0.3 (Jiang et al., 2023) and the Qwen family (Team, 2024; 2025). For performance evaluation, we adopted the widely used LoRA (Hu et al., 2022) and QLoRA (Dettmers et al., 2023) as our baselines to demonstrate DeFT's effectiveness in combining with such PEFT and quantization methods.

**Downstream Tasks**   LLMs are commonly employed for generation and reasoning tasks, which can faithfully well reflect the performance of fine-tuning. Therefore, our experiments mainly focus on arithmetic reasoning and summarizing tasks. For the arithmetic reasoning task, we adopted five widely used datasets covering various difficulties: AddSub (Hosseini et al., 2014), SingleEq (Koncel-Kedziorski et al., 2015), MultiArith (Roy & Roth, 2016), SVAMP (Patel et al., 2021), and GSM8k (Cobbe et al., 2021). Sequences were extracted from each dataset and then composed into the training dataset that has 10000 sequences. The evaluation was performed after the fine-tuning, covering the test set of each dataset, and we used the pass@1 accuracy as the metric. Besides, we followed Hu et al. (2023) and used the scores obtained by GPT-3.5 text-Davinci-003 with Zero-shot Chain-of-Thought (Kojima et al., 2022) as the reference. For the summarizing task, we adopted XSum (Narayan et al., 2018), which is collected from BBC, covering a wide variety of

Table 2: Performance comparison on text summarizing tasks

| Models | Methods | #Params | #Trainable | Rouge1 | Rouge2 | RougeL |
|--------|---------|---------|------------|--------|--------|--------|
| LLaMA-13B | QLoRA | 13.02B | 112M | 42.62 | 17.99 | 34.67 |
|  | +DeFT | 11.06B | 112M | **42.99** | **18.27** | **34.99** |
| Qwen-3 32B | Zero shot | 32.76B | 0M | 18.60 | 3.41 | 13.09 |
|  | QLoRA | 32.76B | 241M | **41.86** | **17.70** | **33.97** |
|  | +DeFT | 28.00B | 241M | 41.39 | 17.17 | 33.78 |

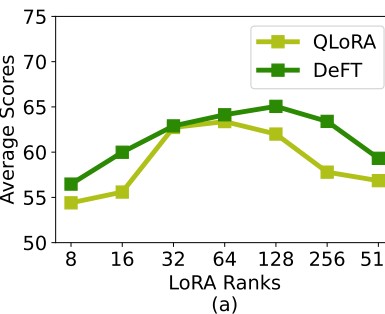
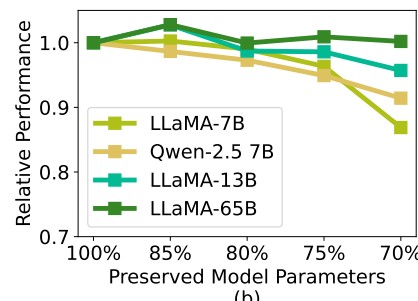

Figure 5: (a) LLaMA-7B's fine-tuning performance on arithmetic reasoning tasks with varying LoRA ranks. (b) DeFT's relative performance against QLoRA under different compression rates.

domains. We use the first 10000 pieces as the training set. The metrics we employed are Rouge scores (Lin, 2004), the most widely used metric to evaluate the similarity between model-generated and manual summaries. For the implementation details, please refer to Appendix C.

### 3.1.1 OVERALL PERFORMANCE

We report the fine-tuning performance on the downstream tasks in Table 1 and Table 2. Combining DeFT with LoRA/QLoRA achieves comparable or even better performance than LoRA/QLoRA across various pre-trained backbones, which demonstrates the effectiveness of DeFT. Additionally, the performance degradation of LoRA+DeFT compared to full fine-tuning is in an acceptable range. For large models such as LLaMA-65B, Qwen-3 32B and LLaMA-3 70B, we do not report their full fine-tuning results, since we have limited training resources.

**Why DeFT Can Outperform LoRA/QLoRA Sometimes?** Large, over-parameterized models may have weights that are noisy or contain components that are not essential for a specific downstream task. The SVD process, guided by our activation-aware search, acts as a form of low-rank regularization. It effectively prunes away the singular components that contribute the least to the feature transformations on the downstream task data (as captured by our calibration data). This removes "distracting" or noisy directions in the weight space, leading to a more stable and task-relevant feature representation. Furthermore, DeFT is not a blind compression but a task-adaptive decomposition. By using calibration data from the downstream task and an outlier-aware importance metric, DeFT prioritizes preserving the weight components that are most critical for the target domain. In contrast, the full QLoRA model retains all parameters, including those that might be optimized for general pre-training knowledge but are less relevant or even counter-productive for the fine-tuning task. Therefore, DeFT isn't just making the model smaller, but making it more "specialized" by concentrating its representational power on the most salient features for the task at hand.

**Varying LoRA Ranks** We present the performance comparison under different LoRA ranks between QLoRA and DeFT on the reasoning tasks. Specifically, we used LLaMA-7B as the backbone and set the ratio of model parameters of DeFT to 85%. Results are reported in Figure 5(a). Comparing with QLoRA, DeFT consistently achieves competitive or better performance under different LoRA ranks, while benefiting from fewer model parameters.

**Varying Compression Rates** Figure 5(b) presents DeFT's relative performance against QLoRA under different compression rates, where the "100%" represents QLoRA's performance. We can

Table 3: Ablation study.

| Methods | PPL (Wiki) | PPL (Train) | Scores |
|---|---|---|---|
| DeFT | 16.2646 | **2.5731** | **62.90** |
| w/ Vanilla Init | 16.2646 | 2.5731 | 62.52 |
| w/o Importance | 15.7393 | 2.5711[1] | 61.86 |
| w/o Search | **14.0243** | 2.6526 | 57.60 |

[1]Unaligned comparison due to change of the coefficient

observe a slight performance improvement at a low compression rate. When the compression rate increases, the performance drops, especially for smaller foundation models. However, this phenomenon gets alleviated when scaling up the model size. For LLaMA-65B, DeFT achieves similar fine-tuning performance against QLoRA when preserving 70% foundation model parameters, and its performance drops 2.2% compared to QLoRA when only preserving 55% parameters (For the details, please refer to Appendix D.4).

### 3.2 In-depth Analysis of DeFT

#### 3.2.1 Ablation Study

To validate the effectiveness of each component in DeFT, we carried out an ablation study on the reasoning tasks using LLaMA-7B as the backbone. We preserve 85% model parameters of DeFT and the results are shown in Table 3, where we respectively disabled the LoRA initialization using the tails of the truncated singular values, the outlier weighted layer importance, and the search for layerwise truncation position.

As shown in Table 3, we compared the fine-tuning performance and also the perplexity of the compressed models on the Wikitext dataset. We can notice a clear discrepancy between the perplexity and fine-tuning performance, where lower perplexity does not indicate better performance. To explore the correlation between fine-tuning performance and the reconstruction error, we evaluated the reconstruction error by computing the compressed model's perplexity on the training set. The result is consistent with our presumption, *i.e.,* lower reconstruction error leads to better fine-tuning performance. The ablation study clearly demonstrates the effectiveness of the proposed layerwise importance aware fine-grained compression for fine-tuning.

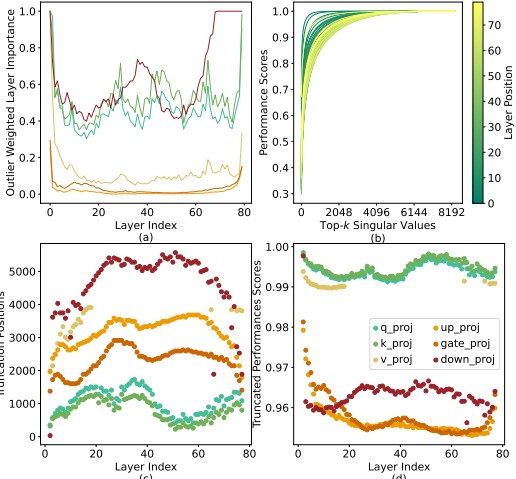

Figure 6: (a) Normalized outlier weighted layer importance (into [0, 1]). (b) Variation of performance score for each "v_proj" layer, where the score is normalized into [0, 1]. (c) Truncation positions of model decomposition. (d) Corresponding performance score to the truncation positions.

#### 3.2.2 Module Sensitivity

DeFT performs a search to determine truncation positions according to equation 6, which considers innate differences among layers. To better illustrate this concept, we explore DeFT which retains 60% of the foundation model parameters of LLaMA-65B (the hidden size is 8192) to showcase the layer difference and how it affects truncation positions. The results are shown in Figure 6. As presented in Figure 6(a) and 6(b), we can observe variations in the component-wise importance and layerwise performance scores, which straightforwardly lead to diverse truncation patterns. For instance, the outlier weighted layer importance of "v_proj" is small for most layers, but "v_proj" in most of the deeper layers has not been decomposed. This is because small changes of the truncation position could lead to dramatic performance score drops for the deeper layers (see Figure 6(b)). DeFT thus selects other components that allow more aggressive truncation positions for decom-

position. The final truncation positions and their corresponding performance scores are shown in Figure 6(c) and Figure 6(d), respectively.

### 3.2.3 EFFICIENCY IMPROVEMENT

One efficiency bottleneck for fine-tuning LLMs with billions of model parameters is data movement. Reducing memory footprint can significantly improve the utilization of high-bandwidth memory I/O, curtailing fine-tuning time expense. DeFT improves both the memory efficiency and end-to-end computation efficiency (including the model decomposition cost and the search cost) for fine-tuning, as shown in Table 4. Compared with QLoRA, DeFT with a compression rate of 55% reduces 22.4%-36.1% memory footprint, improves the throughput by 31.6%-50.6%, and reduces 24.0%-33.6% overall training time under different batch sizes. Furthermore, with a batch size of two, DeFT allows extremely large LLM fine-tuning on resource-constrained devices. For instance, LLaMA-65B, with 55% of its parameters preserved, can be fine-tuned on an NVIDIA RTX4090 with 24GB memory, completing the fine-tuning process in about 15.9 hours.

Table 4: End-to-end fine-tuning efficiency comparison on LLaMA-65B using an NVIDIA A800.

| Method | Batch size | Memory[1] | Throughput[2] | Cost[3] |
|---|---|---|---|---|
| Full FT | 1 | 530.0[4] | - | - |
| QLoRA | 1 | 35.95 | 136.2 | 950.9 |
| | 2 | 36.34 | 214.5 | 637.9 |
| | 4 | 38.76 | 300.4 | 478.2 |
| | 8 | 43.62 | 366.9 | 410.3 |
| | 16 | 52.72 | 411.8 | 383.0 |
| +DeFT | 1 | $22.99_{\downarrow 36.1\%}$ | $179.3_{\uparrow 31.6\%}$ | $722.5_{\downarrow 24.0\%}$ |
| | 2 | $23.37_{\downarrow 35.7\%}$ | $322.3_{\uparrow 50.3\%}$ | $424.5_{\downarrow 33.5\%}$ |
| | 4 | $25.78_{\downarrow 33.5\%}$ | $452.3_{\uparrow 50.6\%}$ | $317.6_{\downarrow 33.6\%}$ |
| | 8 | $33.87_{\downarrow 22.4\%}$ | $546.7_{\uparrow 49.0\%}$ | $275.3_{\downarrow 32.9\%}$ |
| | 16 | $40.34_{\downarrow 23.5\%}$ | $617.3_{\uparrow 49.9\%}$ | $255.5_{\downarrow 33.3\%}$ |

[1]Gigabytes (GB), [2]Tokens/sec, [3]Minutes, [4]Estimated

### 3.2.4 IMPACT OF OUTLIER WEIGHTED LAYER IMPORTANCE

Here, we use LLaMA-7B with DeFT preserving 85% model parameters to explore the impact of $\eta$ on the fine-tuning performance. Results on the arithmetic reasoning tasks are reported in Table 5. There is a clear performance gap when enabling the scaling coefficient, and the performance gradually increases with $\eta$ getting larger. This demonstrates the assumption in our motivation, *i.e.,* evenly compressing all the layers under a preset compression rate overlooks the varying compression sensitivity of different layers.

Table 5: Fine-tuning performance under different $\eta$.

| $\eta$ | 0 | 0.1 | 0.5 | 1 |
|---|---|---|---|---|
| | 61.86 | 58.84 | 60.13 | 62.9 |

### 3.2.5 INTEGRATION WITH ANOTHER QUANTIZATION METHOD LOFTQ

DeFT is a plug-and-play method which can be integrated with PEFT methods (e.g., LoRA) and quantization methods (e.g., QLoRA) to further improve fine-tuning efficiency while matching their performance. Our main experiments in Table 1 select LoRA and QLoRA as two representative methods to combine with DeFT to demonstrate DeFT's effectiveness. However, DeFT can combine with other variants of LoRA and QLoRA. Here, we present the results of combining DeFT with another quantization-based method LoftQ (Li et al., 2024). Specifically, as the open-source codes of LoftQ currently do not support Qwen models, we use LLaMA-2 13B as the backbone to conduct the experiments on the arithmetic reasoning tasks (other settings are the same as in Table 1), and the results are shown in Table 6. The results reveal that beyond LoRA and QLoRA, DeFT is also effective when combining with LoftQ, indicating its good compatibility.

Table 6: Combining DeFT with LoftQ on the arithmetic reasoning tasks using LLaMA-2 13B.

| Methods | AddSub | SingleEq | MultiArith | SVAMP | GSM8k | Avg. |
|---|---|---|---|---|---|---|
| LoftQ | 86.58 | 89.96 | 85.71 | 67.80 | 50.19 | 76.05 |
| LoftQ + DeFT | 87.09 | 92.32 | 83.61 | 67.50 | 47.23 | 75.55 |

## 4 RELATED WORKS

**Parameter-efficient fine-tuning**   PEFT methods can be roughly categorized into the following few types: adapter-based methods (Houlsby et al., 2019; Hu et al., 2023; He et al., 2022), masking-based methods (Guo et al., 2021; Zaken et al., 2022), LoRA-based methods, and Prompt Tuning (Li & Liang, 2021; Liu et al., 2022). Among these PEFT methods, LoRA (Hu et al., 2022) proposes to freeze the pre-trained model and only optimize the newly added low-rank matrices. QLoRA (Dettmers et al., 2023) enhances LoRA by quantizing the pre-trained model into 4-bit precision and utilizing paged optimizers to manage memory spikes. Additionally, quantization-aware fine-tuning is receiving more and more attention and proves to be a practical way to incorporate quantization into model fine-tuning (Li et al., 2024; Xu et al., 2024; Guo et al., 2024).

**Model Decomposition for LLM Inference**   Considerable efforts have been devoted to studying activation-aware model decomposition. It mitigates reconstruction errors brought by vanilla truncated SVD's failure of capturing data distribution (Yuan et al., 2023; Yu & Wu, 2023; Kaushal et al., 2023; Wang et al., 2024). As for truncation position selection, some propose to adopt uniform settings in order to get lower perplexity (Wang et al., 2024), while others try to find the most appropriate configurations for each layer (Yuan et al., 2023; Ji et al., 2024). Despite these considerable efforts of model decomposition for LLM inference, its potential for LLM fine-tuning remains unexplored. In this paper, we propose the first work based on the *decomposition then fine-tuning* paradigm.

## 5 CONCLUSION

In this paper, we introduce a novel method DeFT based on the *decomposition then fine-tuning* paradigm for LLMs. DeFT is empowered with fine-grained foundation model decomposition by an efficient layer importance aware search algorithm. It effectively reduces the number of foundation model parameters during fine-tuning while maintaining the model quality. Besides, DeFT is feasible to incorporate with PEFT and quantization. Experimental results show that DeFT achieves comparable performance or even outperforms the baselines on the arithmetic reasoning and summarizing tasks, while improving both memory and computation efficiency. Impressively, DeFT enables fine-tuning a 65B model on a consumer GPU without using offloading strategies, demonstrating its significant practical value in memory-constrained scenarios.

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

## A DISCLOSE OF LLM USAGE

We only use LLMs to polish our writing, *e.g.,* grammar checking. We do not use LLMs to directly generate the content of this paper.

# B  Layer Importance Aware Truncation Position Search Algorithm

Given Layer set $L$, memory budget $\mathcal{B}$, granularity $G$, layer importance weights $\alpha_l$, performance score function $f$, memory function $g$, and scaling coefficient $\eta$, Algorithm 1 starts with initializing the truncation positions $\theta_l$ to the smallest value such that the performance score $f(\theta_l)$ is at least 0.999 (cf. Line 1). Then the loop continues until the total memory consumption of all layers is within the budget $\mathcal{B}$ (cf. Line 2). For each layer, it computes the loss if the truncation positions are reduced by granularity $G$. The loss combines the change in performance score and memory, weighted by the layer importance (cf. Line 3-11). Subsequently, it selects the layer $l^*$ whose truncation reduction causes the least performance loss per memory saved (cf. Line 12), and reduces the truncation position for layer $l^*$ by $G$ (cf.Line 13). Finally, it returns the final truncation positions $\theta$ (cf. Line 16).

---

**Algorithm 1** Layer Importance Aware Truncation Position Search

---

**Require:** Layers $L$, memory budget $\mathcal{B}$, granularity $G$, layer importance weights $\alpha_l$, performance score function $f$, memory function $g$, scaling coefficient $\eta$

**Ensure:** Truncation positions $\theta_l$ for all $l \in L$

1: Initialize $\theta_l \leftarrow \mathrm{argmax}_r\{f(r) \geq 0.999\}$ for all $l \in L$
2: **while** $\sum_{l \in L} g(\theta_l) > \mathcal{B}$ **do**
3:    **for** each layer $l \in L$ **do**
4:       **if** $\theta_l - G \geq 0$ **then**
5:          $\Delta f_l \leftarrow f(\theta_l) - f(\theta_l - G)$
6:          $\Delta g_l \leftarrow g(\theta_l) - g(\theta_l - G)$
7:          $\mathrm{loss}_l \leftarrow (1 + \eta\alpha_l) \cdot \frac{\Delta f_l}{\Delta g_l}$
8:       **else**
9:          $\mathrm{loss}_l \leftarrow \infty$
10:       **end if**
11:    **end for**
12:    $l^* \leftarrow \mathrm{argmin}_{l \in L} \mathrm{loss}_l$
13:    $\theta_{l^*} \leftarrow \theta_{l^*} - G$
14: **end while**
15:
16: **return** $\theta$

---

# C  Implementation Details

To ensure fair and reproducible experiments, all the baseline implementation and model fine-tuning are based on the publicly available codebases *Huggingface Transformers*[5] and *Huggingface PEFT*[6]. The evaluation procedure is adopted from the publicly available evaluation suite (Hu et al., 2023).

**Hyperparameters**  For LoRA, QLoRA and DeFT, we selected the learning rate from {1e-4, 3e-4, 5e-4}, and set the batch size to 16, the LoRA rank $r$ to 32 with a coefficient of 16. We used the AdamW (Loshchilov & Hutter, 2019) optimizer with default configurations, where beta1 was set to 0.9 and beta2 to 0.999. For full fine-tuning, the learning rate was selected from {5e-6, 1e-5, 2e-5, 5e-5}, and other settings remained the same. For DeFT, we adopted $\eta = 1$ from $\eta \in \{0, 0.1, 0.5, 1.0\}$, and $T$ in layer importance modeling was set to 5 following Yin et al. (2024). Models were evaluated on the test set after 3 epochs of fine-tuning. To make the best use of NVIDIA hardware[7], we set the granularity $G$ to 32. Additionally, the calibration data for DeFT contains 256 samples.

**Model Loading/Quantization Optimization in DeFT**  For model loading in existing model compression techniques, the Huggingface Transformers library first loads all model weights into the

---

[5]https://github.com/huggingface/transformers
[6]https://github.com/huggingface/peft
[7]https://docs.nvidia.com/cuda/cublas/index.html#tensor-core-usage

CPU memory and then calls the quantizer to do the quantization. Then it substitutes pre-trained weights with compressed weights. This is impractical for GPUs that have limited memory, *e.g.,* loading LLaMA-65B on an NVIDIA RTX4090 (24GB).

To mitigate the gap, we further optimize this part in DeFT. With modifying the Huggingface Transformers library, DeFT uses its search results to prevent corresponding weights from being processed in advance. It can directly load and quantize the decomposed weights, *i.e.,* singular vectors, and thus successfully loads LLaMA-65B on an NVIDIA RTX4090.

**Prompts and Training Examples** Supervised fine-tuning engaged in this paper adopted two kinds of prompts. For the arithmetic reasoning tasks, we adopted the prompt format from Alpaca:

> *Below is an instruction that describes a task. Write a response that appropriately completes the request.*
>
> *### Instruction: {instruction}*
>
> *### Response: {response}*

LLMs fine-tuned with arithmetic reasoning datasets are trained to learn how to use chain-of-thoughts to perform reasoning. Here is an example from the training data:

> *Below is an instruction that describes a task. Write a response that appropriately completes the request.*
> *### Instruction:*
> *Wyatt's mother gave him $74 to go to the store. Wyatt bought 5 loaves of bread and 4 cartons of orange juice. Each loaf of bread cost $5 and each carton of orange juice cost $2. How much money does Wyatt have left?*
> *### Response:*
> *Step 1: Find the total cost of the bread by multiplying the cost per loaf by the number of loaves: 5 loaves x $5/loaf = $25.*
> *Step 2: Find the total cost of the orange juice by multiplying the cost per carton by the number of cartons: 4 cartons x $2/carton = $8.*
> *Step 3: Find the total cost of the items by adding the cost of the bread and the cost of the orange juice: $25 + $8 = $33.*
> *Step 4: Subtract the total cost of the items from the amount of money Wyatt started with: $74 - $33 = $41.*
> *Therefore, Wyatt has $41 left.*

For the text summarizing tasks, we adopted a straightforward prompt to fine-tune LLMs to complete this downstream task, which is listed as follows:

> *original text: {text}*
>
> *summary: {summary}*

Here is an example from the training data:

> *original text: Veronica Vanessa Chango-Alverez, 31, was killed and another man injured when an Audi A3 struck them in Streatham High Road at 05:30 GMT on Saturday. Ten minutes before the crash the car was in London Road, Croydon, when a Volkswagen Passat collided with a tree. Police want to trace Nathan Davis, 27, who they say has links to the Audi. The car was abandoned at the scene. Ms Chango-Alverez died from multiple injuries, a post-mortem examination found. No arrests have been made as yet, police said. Ms Chango-Alverez was staying at her mother's home in Streatham High Road. She was born in Ecuador and had lived in London for 13 years, BBC London reporter Gareth Furby said. At the*

Table 7: More experimental results on the arithmetic reasoning tasks.

| Models | Methods | #Params | #Trainable | AddSub | SingleEq | MultiArith | SVAMP | GSM8k | Avg. |
|---|---|---|---|---|---|---|---|---|---|
| OPT-6.7B | Full FT | 6.66B | 6.66B | 58.73 | 55.51 | 55.88 | 28.00 | 11.68 | **41.96** |
| | QLoRA | 6.66B | 75M | 58.48 | 55.32 | 50.84 | 26.50 | 13.12 | 40.85 |
| | +DeFT | 6.17B | 75M | 57.98 | 53.74 | 51.68 | 28.30 | 12.21 | 40.99 |
| LLaMA-13B | Full FT | 13.02B | 13.02B | 85.06 | 85.43 | 79.41 | 62.90 | 43.22 | **71.20** |
| | QLoRA | 13.02B | 112M | 82.03 | 84.05 | 78.43 | 57.97 | 41.57 | 68.81 |
| | +DeFT | 11.06B | 112M | 87.43 | 87.99 | 79.41 | 60.30 | 38.41 | 70.71 |
| LLaMA-33B | QLoRA | 32.53B | 218M | 86.08 | 90.49 | 85.01 | 65.30 | 53.05 | 75.98 |
| | +DeFT | 27.65B | 218M | 89.20 | 91.27 | 83.75 | 67.10 | 51.10 | **76.49** |
| Mistral-7B v0.3 | Zero shot | 7.25B | 0M | 79.24 | 74.80 | 64.29 | 66.90 | 47.23 | 66.49 |
| | Full FT | 7.25B | 7.25B | 89.11 | 92.72 | 87.82 | 69.90 | 54.06 | 78.72 |
| | QLoRA | 7.25B | 75M | 88.61 | 94.29 | 88.66 | 70.30 | 54.44 | **79.26** |
| | +DeFT | 6.20B | 75M | 88.86 | 93.11 | 86.13 | 67.70 | 53.90 | 77.94 |
| | LoRA | 7.25B | 75M | 89.62 | 92.91 | 87.82 | 69.90 | 52.99 | 78.65 |
| | +DeFT | 6.20B | 75M | 86.33 | 93.70 | 87.39 | 68.30 | 55.57 | 78.26 |

*time of the crash, she was on her way to work in a hotel. The remains of the bus stop, which was extensively damaged in the crash, have been removed. Flowers have been left at the site in tribute to the victim. A statement from her brother Kevin Raul Chango-Alverez said: "My family has had its heart torn out, at this Christmas time, we will never be the same again. "On Friday night we were together as a family with Veronica meeting her newly born nephew and preparing for Christmas. "I last saw her alive as she left to go to work on Saturday morning, but moments later I was holding her hand as she passed away in the street." Describing the crash as "horrific" Det Insp Gordon Wallace, said: "The family are devastated. The memory of this senseless death will be with them each time they leave their home. "The driver fled the scene abandoning the grey Audi, which was extensively damaged. "We are looking to speak to Mr Nathan Davis in relation to this collision." The 51-year-old man injured at the bus stop remains in a critical condition in hospital while the condition of the 29-year-old driver of the Volkswagen is now stable.*

*summary: A man with links to a car that was involved in a fatal bus stop crash in south London is being sought by police.*

Models are fine-tuned based on the ground-truth, *i.e.,* "{*response*}" and "{*summary*}".

# D ADDITIONAL EXPERIMENTAL RESULTS

## D.1 ADDITIONAL RESULTS OF FINE-TUNING PERFORMANCE

In this section, we present additional experimental results of different pre-trained backbones on the arithmetic reasoning tasks, as detailed in Table 7. It is shown that DeFT consistently delivers competitive performance alongside QLoRA across various pretrained backbones, *e.g.,* OPT, LLaMA and Mistral. This consistency further underscores the effectiveness of DeFT.

Beyond the arithmetic reasoning tasks, here we present additional results on another more challenging task MATH (Hendrycks et al.). We conduct the experiments using two different models, i.e., LLaMA-2 13B and Qwen-2.5 7B. We fine-tune the model on the training data for three epochs and evaluate on the MATH-500 test set. The accuracy@1 results are shown in Table 8. These results further demonstrate DeFT's effectiveness even on such challenging tasks.

Table 8: Fine-tuning performance on the MATH task.

| Method | LLaMA-2 13B | Qwen-2.5 7B |
|---|---|---|
| QLoRA | 4.80 | 36.3 |
| QLoRA + DeFT | 6.80 | 36.0 |

## D.2 Additional Results of Fine-tuning Efficiency

Here, we present the end-to-end fine-tuning efficiency (including the model decomposition and search cost) comparison on LLaMA-33B using a single NVIDIA RTX4090 GPU, where the ratio of the preserved model parameters of DeFT is set to 75%. The results are presented in Table 9. Compared with QLoRA, DeFT achieves up to 19.6%, 49.0% and 32.4% improvements in terms of memory efficiency, throughput and end-to-end training time, respectively, consistently demonstrating the efficiency benefits of DeFT.

Table 9: End-to-end Fine-tuning efficiency comparison on LLaMA-33B using a single NVIDIA RTX4090 GPU.

| Method | Batch Size | Memory[1] | Throughput[2] | Cost[3] |
|---|---|---|---|---|
| QLoRA | 1 | 19.45 | 146.5 | 884.2 |
| | 2 | 20.56 | 245.1 | 558.1 |
| | 4 | 22.54 | 350.9 | 409.5 |
| +DeFT | 1 | $16.59_{\downarrow14.7\%}$ | $218.3_{\uparrow49.0\%}$ | $597.3_{\downarrow32.4\%}$ |
| | 2 | $16.54_{\downarrow19.6\%}$ | $351.1_{\uparrow43.2\%}$ | $393.2_{\downarrow29.5\%}$ |
| | 4 | $19.26_{\downarrow14.6\%}$ | $488.8_{\uparrow39.3\%}$ | $311.6_{\downarrow31.4\%}$ |

[1]Gigabytes (GB), [2]Tokens/sec, [3]Minutes

## D.3 Impact of the Calibration Data Size

Activation-aware singular value decomposition methods usually require calibration data to capture activation information and reduce decomposition errors. Existing practice tends to construct calibration data out of the pre-training dataset since it seeks to retain generation quality on general tasks. However, for DeFT, we aim to repurpose model decomposition for fine-tuning. Therefore, we construct calibration data from the downstream tasks. To explore how this affects the fine-tuning performance, we use DeFT preserving 85% model parameters on LLaMA-7B to investigate the fine-tuning performance under different sizes of calibration data. The results are reported in Table 10. We notice that with a calibration data size of 256, DeFT achieves the highest score. Therefore, we adopted this in our experiments.

Table 10: Fine-tuning performance under different calibration data sizes.

| #Calibration | 32 | 64 | 128 | 256 |
|---|---|---|---|---|
| Avg. Score | 62.64 | 61.10 | 61.88 | 62.90 |

Additionally, we investigate the impact of different calibration data subsets on the final performance. Specifically, we use Qwen-2.5 7B as the backbone and randomly select the calibration data with three different random seeds. We present the average performance on the arithmetic reasoning tasks, as shown in Table 11. The results reveal that the performance of DeFT is sensitive to the calibration data subset, which is reasonable since the quality of the selected calibration data has an important impact on the model decomposition. For our experiments in this paper, we have fixed the random seed for calibration data selection to eliminate the impact of this factor.

Table 11: Fine-tuning performance under different calibration data subsets.

| Random seed | 42 | 43 | 44 |
|---|---|---|---|
| Avg. Score | 86.42 | 85.65 | 87.11 |

## D.4 Results of 65B Model Varying Compression Rates

Figure 7 presents the performance of DeFT across different preserved model parameters on LLaMA-65B. DeFT outperforms QLoRA when the compression rate is smaller, *i.e.,* more preserved model parameters (85% and 75%). When the model is aggressively compressed, *e.g.,* only preserving 55% model parameters, there is an inevitable performance drop, but the drop range is acceptable, *i.e.,* 2.2%. These results demonstrate the strong performance-efficiency trade-off of DeFT.

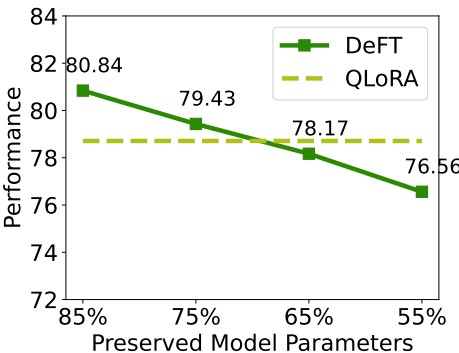

Figure 7: Performance of DeFT across different preserved model parameters on LLaMA-65B.

## D.5 COMPARISON WITH ADAPTER-TUNING METHODS

Here, we compare DeFT with existing widely-used adapter-tuning methods (Houlsby et al., 2019; Pfeiffer et al., 2021). In more detail, we present the performance comparison on the reasoning tasks and text summarizing tasks using LLaMA-7B and LLaMA-13B as the backbones. For the adapter-tuning baselines, we kept the batch size to 16 and conducted a grid search for the learning rate and learning rate scheduler settings from {1e-6, 2e-6, 5e-6, 1e-5, 2e-5} and {*constant*, *cosine*}. Other fine-tuning and evaluation procedures are kept the same for a fair comparison. The results are reported in Table 12. We can see that DeFT significantly outperforms the adapter-tuning methods, especially on the summarizing tasks.

Table 12: Performance comparison with adapter-tuning methods, where scores for reasoning tasks are the average pass@1 accuracy and scores for summarizing tasks are RougeL.

| Task | Model | Full FT | Series[*] | Parallel[*] | QLoRA | DeFT |
|---|---|---|---|---|---|---|
| Reasoning | 7B | 63.98 | 60.64 | 62.53 | 62.73 | 62.90 |
|  | 13B | 71.20 | 70.50 | 65.60 | 68.81 | 70.71 |
| Summarizing | 7B | 34.28 | 25.53 | 25.98 | 33.53 | 34.03 |
|  | 13B | 35.29 | 26.37 | 26.88 | 34.67 | 34.99 |

[*] Adapter-based tuning.

## D.6 PERFORMANCE STABILITY

With model parameters reduced, the decomposed models inevitably suffer from increasing reconstruction error, making their fine-tuning performance less stable, especially for smaller models. We conducted repeated experiments on the reasoning tasks under three different random seeds and computed the standard deviations of the average scores, which are reported in Table 13. For LLaMA-7B/33B, the standard deviations of the average scores increase with the reduction of the model parameters. However, such phenomena are mitigated when it comes to LLaMA-65B. This is because, for smaller foundation models, the reconstruction error could be relatively too large to be compensated, whereas for larger models, the same level of error becomes relatively small due to greater parameter redundancy.

Table 13: Standard deviations of scores on the reasoning tasks

| DeFT | LLaMA-7B | LLaMA-33B | LLaMA-65B |
|---|---|---|---|
| 85% | 0.52 | 0.93 | 0.32 |
| 80% | 0.68 | 1.74 | 0.41 |
| 75% | 1.09 | 2.27 | 0.49 |
| 70% | 2.27 | 2.31 | 0.51 |

Table 14: Performance of full fine-tuning DeFT on the reasoning tasks with Qwen-2.5 7B.

| DeFT | AddSub | SingleEq | MultiArith | SVAMP | GSM8k |
|------|--------|----------|------------|-------|-------|
| + LoRA | **93.16** | **96.06** | **93.28** | **84.70** | **71.42** |
| + Full FT | 85.82 | 90.55 | 87.39 | 72.70 | 57.24 |

## D.7 PERFORMANCE OF INCORPORATING DEFT WITH FULL FINE-TUNING

In addition to combining DeFT with PEFT and quantization methods, one may consider whether DeFT can be combined with full fine-tuning, *i.e.,*, directly fine-tuning the decomposed model. To answer this question, we explore the performance of full fine-tuning DeFT on the reasoning tasks using Qwen-2.5 7B as the backbone, and the results are shown in Table 14. It can be observed that DeFT + LoRA significantly outperforms DeFT + Full fine-tuning.

There are two reasons for this phenomenon. The first one is the inconsistent optimization objective between model decomposition and fine-tuning. For model decomposition, we aim to minimize the compression loss $\|WX - W'X\|_F$, where $W$ is the original pre-trained weight, $W'$ is its low-rank approximation, which is reconstructed based on the decomposed weights, and $X$ is the input. However, for LLM fine-tuning, the goal is to use the downstream task-specific data to maximize the probability of the model to predict the right next token, typically by minimizing the cross-entropy loss. Therefore, if directly fine-tuning the decomposed weights, the fine-tuned weight may no longer be the low-rank approximation of the original weight. Instead, by applying LoRA fine-tuning to the decomposed model, we can keep the decomposed weights frozen and only update the LoRA modules. As such, the decomposed weights are still an approximation to the original weights. The second reason is also mentioned in the work of SVD-LLM (Wang et al., 2024): the derivatives of the decomposed weights are interdependent during the fine-tuning process, where optimization of one matrix may interfere with the optimization of the other, leading to a performance drop. Due to these two reasons, we cannot achieve satisfactory performance by full fine-tuning DeFT.

