# OpenReview forum: "Efficient Fine-tuning with Decomposed Foundation Model"
_ICLR.cc/2026/Conference — Submitted to ICLR 2026_

### Official Review · Reviewer_ZfCd · 2025-10-27

**Soundness:** 3
**Presentation:** 3
**Contribution:** 3
**Rating:** 6
**Confidence:** 4

**Summary:**

The experimental results are a major strength. DeFT not only achieves comparable or even slightly better performance than standard QLoRA/LoRA baselines, but it does so with a significantly smaller model. The headline claim, backed by experiments (Table 4), is that DeFT enables the fine-tuning of a 65B parameter model on a single 24GB consumer GPU (e.g., RTX 4090) without any offloading strategies, reducing memory usage by ~36% and total training time by ~24-33%.

**Strengths:**

The primary strength of this paper is its immense practical value. The ability to fine-tune a 65B (or 70B) model on a single 24GB consumer GPU is a game-changer. It breaks down a major hardware barrier and democratizes access to large-scale fine-tuning for a much wider range of researchers and developers.

DeFT is not just "SVD + LoRA." The core novelty lies in the fine-grained decomposition search algorithm. This algorithm (Eq. 6) is principled, intelligently optimizing a clear trade-off (performance loss vs. memory gain) and weighting it by layer sensitivity. The ablation study (Table 3) compellingly proves that this search is the most critical component of DeFT, as a naive uniform compression (w/o Search) fails dramatically.

**Weaknesses:**

The paper's most surprising finding is that the smaller, compressed +DeFT models often outperform the larger, baseline QLoRA models (e.g., LLaMA-65B, 80.84 vs. 78.71 avg). While the ablation shows what causes this (the search algorithm), it doesn't fully explain why. Is this a regularization effect? Does the SVD process "clean" noisy, over-parameterized weights? A deeper discussion of this "compression-for-a-gain" phenomenon would strengthen the paper.

The method relies on a small set of calibration data (256 samples from the downstream task) to calculate activation statistics for both the SVD whitening and the layer importance ($\alpha_l$). Appendix C.3 shows performance is sensitive to the size of this data. More analysis on its sensitivity to the content (e.g., different random 256-sample sets) would be valuable.

**Questions:**

the finding that +DeFT models (which are smaller) can outperform their baseline QLoRA counterparts is fascinating. What is the authors' primary hypothesis for this? Is it a regularization effect from the low-rank constraint, or does the activation-aware SVD process somehow "clean up" the pre-trained weights in a way that is beneficial for fine-tuning?

What is the one-time pre-processing cost (SVD decomposition + search) for the largest models, LLaMA-65B and LLaMA-3 70B? The 10-minute cost for 7B is excellent, but how does this cost scale with model size?

The use of the SVD-tail to initialize the LoRA adapter is a very clever idea, and Table 3 shows it provides a small but real benefit (62.90 vs. 62.52). Does this imply that the truncated singular vectors (the "discarded" information) are somehow intrinsically well-suited for adaptation, which is exactly what LoRA does? This seems like a powerful insight if true.

---

> ### Author Response · Authors · 2025-11-19
> **Rebuttal by authors**
>
> Thank you for your thoughtful and constructive review. We have carefully addressed each of your concerns in detail. In the following, we provide point-by-point responses to your comments.
>
>
> > W1 & Q1: The paper's most surprising finding is that the smaller, compressed +DeFT models often outperform the larger, baseline QLoRA models (e.g., LLaMA-65B, 80.84 vs. 78.71 avg). While the ablation shows what causes this (the search algorithm), it doesn't fully explain why. Is this a regularization effect? Does the SVD process "clean" noisy, over-parameterized weights? A deeper discussion of this "compression-for-a-gain" phenomenon would strengthen the paper.
>
> Answer: We thank the reviewer for this insightful observation. We have added a dedicated discussion in Section 3.1.1 to elaborate on this, structured around the following points for the "compression-for-a-gain" phenomenon:
>
> - Implicit Low-Rank Regularization and Noise Reduction: Large, over-parameterized models may have weights that are noisy or contain components that are not essential for a specific downstream task. **The SVD process, guided by our activation-aware search, acts as a form of low-rank regularization. It effectively prunes away the singular components that contribute the least to the feature transformations on the downstream task data** (as captured by our calibration data). This removes "distracting" or noisy directions in the weight space, leading to a more stable and task-relevant feature representation.
> - Task-Relevant Feature Preservation: Our method is not a blind compression; it is a task-adaptive decomposition. By using calibration data from the downstream task and an outlier-aware importance metric, DeFT prioritizes preserving the weight components that are most critical for the target domain. In contrast, the full QLoRA model retains all parameters, including those that might be optimized for general pre-training knowledge but are less relevant or even counter-productive for the fine-tuning task. Therefore, **DeFT isn't just making the model smaller; it's making it more "specialized" by concentrating its representational power on the most salient features for the task at hand.**
>
>
> > W2: The method relies on a small set of calibration data (256 samples from the downstream task) to calculate activation statistics for both the SVD whitening and the layer importance (\$a_l\$). Appendix C.3 shows performance is sensitive to the size of this data. More analysis on its sensitivity to the content (e.g., different random 256-sample sets) would be valuable.
>
> Answer: Thanks for your suggestion. We have added additional experiments to analysis DeFT’s sensitivity to different random 256-sample sets of calibration data. Specifically, we use Qwen-2.5 7B as the backbone and randomly select the calibration data with three different random seeds. The average performance on the arithmetic reasoning tasks is shown below.
> | Random seed | Avg. score |
> | - | :-: |
> | 42 | 86.42 |
> | 43 | 85.65 |
> | 44 | 87.11 |
>
> The results reveal that the performance of DeFT is sensitive to the calibration data subset, which is reasonable since the quality of the selected calibration data has an important impact on the model decomposition. **For our experiments in this paper (e.g., Table 1-3, Figure 5-6), we have fixed the random seed for calibration data selection to eliminate the impact of this factor.** We have added this analysis in the revised PDF (Appendix C.3).
>
> > Q2: What is the one-time pre-processing cost (SVD decomposition + search) for the largest models, LLaMA-65B and LLaMA-3 70B? The 10-minute cost for 7B is excellent, but how does this cost scale with model size?
>
> Answer: The cost of one-time SVD decomposition and search is about 1.5 hours for LLaMA-65B and LLaMA-3 70B. It is an acceptable cost for these extremely large models.
>
> > Q3: The use of the SVD-tail to initialize the LoRA adapter is a very clever idea, and Table 3 shows it provides a small but real benefit (62.90 vs. 62.52). Does this imply that the truncated singular vectors (the "discarded" information) are somehow intrinsically well-suited for adaptation, which is exactly what LoRA does? This seems like a powerful insight if true.
>
> Answer: We thank the reviewer for this insightful observation and positive feedback on our initialization strategy. The reviewer has correctly identified a key intuition behind our design: The core premise of LoRA is that the weight change during fine-tuning has a low "intrinsic rank." **Our method extends this by hypothesizing that the residual information not captured by the primary SVD truncation also resides in a low-rank space.** Therefore, it is natural to use this pre-existing, structured low-rank signal (the SVD tail) to initialize the LoRA matrices, rather than initializing them to zero. This effectively seeds the adapter with a prior that is mathematically derived from the original weight matrix and the calibration data.

---

### Official Review · Reviewer_mCwb · 2025-11-03

**Soundness:** 3
**Presentation:** 1
**Contribution:** 2
**Rating:** 4
**Confidence:** 3

**Summary:**

This paper proposes DeFT (Decomposition-then-Fine-Tuning), a framework that leverages model decomposition to improve the efficiency of fine-tuning large language models (LLMs). The core idea is to decompose the foundation model before fine-tuning, thereby reducing parameter count and memory usage while maintaining strong performance. Specifically, DeFT applies activation-aware singular value decomposition (SVD) with whitening to capture input-distribution information and then performs a layer-importance-aware search to determine optimal truncation points based on outlier-weighted sensitivity scores. The truncated singular values initialize the LoRA modules, while the remaining components reconstruct the frozen base model. During training, only the lightweight LoRA parameters are updated.

**Strengths:**

- The proposed approach is novel compared with existing low-rank adaptation methods.
- DeFT demonstrates strong empirical performance across multiple LLMs (e.g., LLaMA, Qwen) and task domains.
- Results are evaluated with multiple metrics such as memory usage and accuracy, providing a comprehensive empirical assessment.

**Weaknesses:**

- The method relies heavily on matrix decomposition and search, which may reduce conceptual clarity.
- The paper’s overall presentation and writing quality could be significantly improved; it currently reads as if prepared under time pressure and would benefit from further polishing.
- Some technical details are insufficiently explained, making it difficult for others to reproduce the results accurately.

**Questions:**

- Definition of WS (L98/L135, Figure 2):
What exactly does “WS” denote? How does the method determine whether a pre-trained weight qualifies as WS and should be decomposed using SVD?

- Computational Overhead:
The activation-aware compression loss involves truncating singular values per weight matrix and position. Does this introduce additional computational overhead, especially when evaluated multiple times for each layer?

- Layer Position (Figure 3b):
What does “Layer position” specifically represent in the figure? Is it the depth within the transformer stack or an abstract ordering metric?

- Cache Mechanism (L261):
Could the authors elaborate on the “cache mechanism” mentioned in L261? How does it function to reduce the computational cost or memory overhead during decomposition or fine-tuning?

- Notation of A, B in Figure 2:
What do A* and B* represent? Are they LoRA adapters, or do they differ from the A and B in the same figure? Clarifying their relationship would help readers follow the architecture more easily. Similar to this, the paper should identify all notations and make them easier to be followed by readers.

- Source of Memory Savings:
Since DeFT still initializes LoRA modules using truncated singular values, it is unclear where the training-time memory reduction primarily comes from. The paper should explicitly describe whether the savings stem from reduced activation storage, frozen base layers, or smaller gradient tensors.

---

> ### Author Response · Authors · 2025-11-19
> **Rebuttal by authors**
>
> Thank you for your thoughtful and constructive review. We have carefully addressed each of your concerns in detail. In the following, we provide point-by-point responses to your comments.
>
> > W1 - W3: presentation issues & insufficient explain of technical details.
>
> Answer: We apologize for the presentation issues. We have improved the clarity of our paper. For example, we have improved the notations and added an algorithmic pseudocode in Section 2 and Appendix B.
>
> > Q1: Definition of WS (L98/L135, Figure 2): What exactly does “WS” denote? How does the method determine whether a pre-trained weight qualifies as WS and should be decomposed using SVD?
>
> Answer: \$S\$ is the Cholesky decomposition of \$ XX^T\$, where \$X\$ is the input feature. \$W\$ is a pretrained weight. We decompose the model weights with SVD since the mathematical guarantee of SVD makes it easy to estimate fine-tuning performance through theoretical compression loss. SVD is performed on \$WS\$ rather than \$W\$, aiming to leverage activation to mitigate reconstruction error brought by outliers (Line 136-141). We have revised Section 2 to improve clarity.
>
> > Q2: Computational Overhead: The activation-aware compression loss involves truncating singular values per weight matrix and position. Does this introduce additional computational overhead, especially when evaluated multiple times for each layer?
>
> Answer: **Our truncation position selection algorithm is an efficient approximate algorithm, which can be finished in a few seconds** (as introduced in Line 255-257). In contrast, the original optimization problem is a typical integer programming problem with a vast solution space, where the exhaustive search is infeasible.
>
> > Q3: Layer Position (Figure 3b): What does “Layer position” specifically represent in the figure? Is it the depth within the transformer stack or an abstract ordering metric?
>
> Answer: “Layer position” in Figure 3b represents the i-th transformer block. We have clarified this in the revised PDF.
>
> > Q4: Cache Mechanism (L261): Could the authors elaborate on the “cache mechanism” mentioned in L261? How does it function to reduce the computational cost or memory overhead during decomposition or fine-tuning?
>
> Answer: DeFT processes the one-time SVD decomposition for each model offline and caches the results to the disk, and the results can be reused for the search algorithm to find the best truncation positions of different compression rates. That means, for different settings of compression rates, we do not need to repeat the SVD decomposition process, saving notable computation cost. The search results can also be cached to the disks for reuse. We have improved the clarity of this part in the revised PDF (Section 2.6).
>
> > Q5: Notation of A, B in Figure 2: What do A* and B* represent? Are they LoRA adapters, or do they differ from the A and B in the same figure? Clarifying their relationship would help readers follow the architecture more easily. Similar to this, the paper should identify all notations and make them easier to be followed by readers.
>
> Answer: Sorry for the confusion. A* and B* represent LoRA adapters, and they are initialized by A and B, respectively. We have improved the notations in Section 2.
>
> > Q6: Source of Memory Savings: Since DeFT still initializes LoRA modules using truncated singular values, it is unclear where the training-time memory reduction primarily comes from. The paper should explicitly describe whether the savings stem from reduced activation storage, frozen base layers, or smaller gradient tensors.
>
> Answer: **The memory savings of DeFT come from the reduced number of model parameters of the frozen foundation model.** As introduced in Line 047-051, PEFT and quantization-aware fine-tuning methods cannot reduce the number of foundation model parameters. However, memory consumption by tens of billions of model parameters poses considerable challenges to fine-tuning, especially in memory-constraint scenarios. DeFT can be integrated with PEFT and quantization-aware fine-tuning to further improve fine-tuning efficiency
>
> We hope we have addressed your concerns. If you have further questions, we are happy to address.

---

### Official Review · Reviewer_LY5k · 2025-11-08

**Soundness:** 2
**Presentation:** 3
**Contribution:** 2
**Rating:** 4
**Confidence:** 4

**Summary:**

The authors introduce decomposition then fine-tuning (DeFT) paradigm, which decomposes the foundation model and then reduces the number of model parameters during fine-tuning, thus enhancing both memory-efficiency and computation-efficiency for
fine-tuning. In addition, DeFT can be seamlessly combined with PEFT and quantization methods. As a result, DeFT allows for fine-tuning of a 65B model on a commercial off-the-shelf GPU with 24GB of memory.

**Strengths:**

- The authors empirically show that DeFT can be integrated with PEFT and quantization methods (e.g., LoRA, QLoRA).
- The authors demonstrate that DeFT can save GPU memory usage by up to 40%, thereby allowing for fine-tuning of a 65B model on a commercial off-the-shelf GPU with 24GB memory.
- The authors conducted extensive experiments for various LLMs including recent LLMs (e.g., Qwen3).

**Weaknesses:**

- The baselines (LoRA and QLoRA) are too limited to verify the effectiveness of DeFT. There are so many variants of LoRA and QLoRA (e.g., DoRA, LoftQ), but it is hard to determine whether DeFT is also effective for those variants or not at this moment.
- Given that the tasks ranging from AddSub to GSM8k are relatively easy, it is difficult to justify the efficacy of DeFT. It would be more beneficial if the authors use more challenging tasks such as MATH.

**Questions:**

N.A.

---

> ### Author Response · Authors · 2025-11-19
> **Rebuttal by authors**
>
> Thank you for your thoughtful and constructive review. We have carefully addressed each of your concerns in detail. In the following, we provide point-by-point responses to your comments.
>
> > W1: The baselines (LoRA and QLoRA) are too limited to verify the effectiveness of DeFT. There are so many variants of LoRA and QLoRA (e.g., DoRA, LoftQ), but it is hard to determine whether DeFT is also effective for those variants or not at this moment.
>
> Answer: We have added additional experiments to combine DeFT with a variant of QLoRA, i.e., LoftQ. Specifically, as the open-source codes of LoftQ currently do not support Qwen models, we use LLaMA-2 13B as backbone to conduct the experiments on the arithmetic reasoning tasks, and the results are shown below:
>
> | Method | AddSub | SingleEq | MultiArith | SVAMP | GSM8k | Avg |
> | - | :-: | :-: | :-: | :-: | :-: | :-: |
> | LoftQ | 86.58 | 89.96 | 85.71 | 67.80 | 50.19 | 76.05 |
> | LoftQ + DeFT | 87.09 | 92.32 | 83.61 | 67.50 | 47.23 | 75.55 |
>
> **Beyond LoRA and QLoRA, DeFT is also effective when combining with LoftQ, indicating its good compatibility.**
>
>
> > W2: Given that the tasks ranging from AddSub to GSM8k are relatively easy, it is difficult to justify the efficacy of DeFT. It would be more beneficial if the authors use more challenging tasks such as MATH.
>
> Answer: Following your suggestion, we have conducted additional experiments on the MATH task using two different models, i.e., LLaMA-2 13B and Qwen-2.5 7B. We perform fine-tuning for 3 epochs and present the acc@1 results:
>
> | Method | LLaMA-2 13B | Qwen-2.5 7B |
> | - | :-: | :-: |
> | QLoRA | 4.80 | 36.3 |
> | QLoRA + DeFT | 6.80 | 36.0 |
>
> **The results further demonstrate DeFT’s effectiveness even on such challenging tasks.**
>
> We hope we have addressed your concerns. If you have further questions, we are happy to address.

---

### Official Review · Reviewer_QxzS · 2025-11-08

**Soundness:** 2
**Presentation:** 2
**Contribution:** 1
**Rating:** 2
**Confidence:** 4

**Summary:**

This paper presents DeFT for memory efficiency of finetuning. The core component is (1) SVD based weight decomposition on $W S$ where $S = X X^\top$ is the empirical covariance matrix built from the calibration data (2) a constrained importance score optimization problem that aims to adaptively select different SVD selection threshold per layer as Eqn 5. It aims to optimize the performance while satisfying the memory efficiency and highest output reconstruction error at the same time. The experiments are conducted on Llama series 1 2 3, QWen 2 and 3, with main baselines as full FT, zeroshot, and QLoRA. The main benchmarks are arithmetic reasoning tasks, text summarization, and wiki perplexity. DeFT matches the performance of QLoRA with 20% memory efficiency savings.

**Strengths:**

- The paper flow is clear and easy to follow even upon the first time of read. Figure 2 provides a clear high-level illustration of DeFT.

- The experiment setup follows the standard approach and well illustrated.

**Weaknesses:**

- **The novelty is limited.** The SVD decomposition is mainly based on Wang et al. [1] and the main technical novelty besides Wang et al. is on the layerwise important weight search process. The iterative search process is a simple alternating optimization and cannot be considered as main novelty. From my perspective, DeFT is a simple if not incremental extension of SVD-LLM at best.

- **The baselines are insufficient**. **DeFT as a weight decomposition method should be compared against alternative pruning/quantization methods also applied to the model weights. There is no such comparison on the main paper besides QLoRA**. I could perceive an alternative better int4 quantization / pruning method also equipped with LoRA finetuning matches DeFT performance and memory efficiency.

- **The memory efficiency besides QLoRA is marginal compared to the extra overhead.** If I understand correctly, DeFT will not save *any* activation memory during finetuning as the decomposed weights are also kept frozen, and the DeFT only saves 20% parameters. From Table 1 and 4, DeFT only save <20% memory besides QLoRA, which I believe should mainly come from loading 20% less master weights per layer. **This is not a major saving if we also consider the overhead of SVD reconstruction.** Specifically, I could perceive offloading 20% more weights to CPU and load twice with 2 reduction can have the same memory efficiency and *much* higher throughput during inference than SVD reconstruction. If the whole memory savings from SVD is less than 30-50%, there will be plentiful alternatives that achieve better tradeoff between memory and compute efficiency during inference.

The second and third weakness are critical (in my opinion this is even more critical than technical novelty as we cannot position DeFT among related works without clear comparison) and I would vote for reject.

Reference:
[1] Xin Wang, Yu Zheng, Zhongwei Wan, and Mi Zhang. SVD-LLM: Truncation-aware Singular Value Decomposition for Large Language Model Compression, April 2024.

**Questions:**

Although the high-level illustration of DeFT is generally clear, the iterative search in sec 2.5 is not clear/detailed enough to show the entire search process. An algorithmic pseudocode illustrating such search formally is needed.

Other questions are listed in weakness above.

**Details Of Ethics Concerns:**

Not needed

---

> ### Author Response · Authors · 2025-11-19
> **Rebuttal by authors**
>
> Thank you for your thoughtful and constructive review. Below, we want to clarify some misunderstandings and respond to your suggestions point-by-point.
>
> > W1: The novelty is limited. The SVD decomposition is mainly based on Wang et al. [1] and the main technical novelty besides Wang et al. is on the layerwise important weight search process. The iterative search process is a simple alternating optimization and cannot be considered as main novelty.
>
> Answer: DeFT's core contribution includes a novel conceptual and technical paradigm shift: repurposing model decomposition for fine-tuning, which necessitates several key innovations:
>
> - **We are the first to formulate the problem of fine-grained model decomposition specifically for fine-tuning.** This is a different goal from inference-time compression.
> - The Search Algorithm is Central to the Contribution: **It is a highly efficient, approximate algorithm tailored for our specific, non-convex integer programming problem.** It is not a generic alternating method but is carefully designed to start from an infeasible point (high performance, high memory) and iteratively move towards the budget constraint hyperplane by making the least damaging truncation at each step (Eq. 6).
> - Selective Model Loading/Quantization: **This allows us to load and quantize only the decomposed singular values, bypassing the need to load the full pre-trained model.** This insists to enable fine-tuning a 65B model on a 24GB GPU, a feat not demonstrated by existing PEFT and 4-bit quantization-aware fine-tuning methods (e.g., QLoRA).
>
> > W2: The baselines are insufficient. DeFT as a weight decomposition method should be compared against alternative pruning/quantization methods also applied to the model weights. There is no such comparison on the main paper besides QLoRA.
>
> Answer: **DeFT is a plug-and-play method** which can be integrated with PEFT methods (e.g., LoRA) and quantization methods (e.g., QLoRA) to further improve memory efficiency while matching their performance. Our experiments select LoRA and QLoRA as two representative methods to combine with DeFT to demonstrate its effectiveness. However, **DeFT can combine with other quantization methods such as LoftQ.**
>
> We have added additional experiments of combining DeFT with LoftQ. Specifically, as the open-source codes of LoftQ currently do not support Qwen models, we use LLaMA-2 13B as the backbone to conduct the experiments on the arithmetic reasoning tasks, and the results are shown below:
>
> | Method | AddSub | SingleEq | MultiArith | SVAMP | GSM8k | Avg |
> | - | :-: | :-: | :-: | :-: | :-: | :-: |
> | LoftQ | 86.58 | 89.96 | 85.71 | 67.80 | 50.19 | 76.05 |
> | LoftQ + DeFT | 87.09 | 92.32 | 83.61 | 67.50 | 47.23 | 75.55 |
>
> **Beyond LoRA and QLoRA, DeFT is also effective when combining with LoftQ, indicating its good compatibility.**
>
> > W3: The memory efficiency besides QLoRA is marginal compared to the extra overhead. If I understand correctly, DeFT will not save any activation memory during finetuning as the decomposed weights are also kept frozen, and the DeFT only saves 20% parameters. From Table 1 and 4, DeFT only save <20% memory besides QLoRA, which I believe should mainly come from loading 20% less master weights per layer. This is not a major saving if we also consider the overhead of SVD reconstruction. Specifically, I could perceive offloading 20% more weights to CPU and load twice with 2 reduction can have the same memory efficiency and much higher throughput during inference than SVD reconstruction.
>
> Answer: As shown in Table 4, compared to QLoRA, **DeFT save 22.4%~36.1% memory consumption** for fine-tuning under different batch sizes, while **reducing the end-to-end fine-tuning time by 24.0%~33.6%.** This is because one efficiency bottleneck for fine-tuning LLMs is data movement. Reducing the memory footprint can significantly improve the utilization of high-bandwidth memory I/O, curtailing fine-tuning time expense. Notably, DeFT allows fine-tuning a 65B model on an NVIDIA RTX4090 with 24GB memory, which can be finished in 15.9 hours. **While offloading can achieve similar memory efficiency, it increases the fine-tuning overhead brought by data movement between CPU and GPU. In comparison, DeFT improves memory efficiency without sacrificing computation efficiency.**
>
> > Q1: Although the high-level illustration of DeFT is generally clear, the iterative search in sec 2.5 is not clear/detailed enough to show the entire search process. An algorithmic pseudocode illustrating such search formally is needed.
>
> Answer: Thank you for your suggestion. We have added an algorithmic pseudocode and its description in Appendix B.
>
> Thank you again for your valuable time. We sincerely hope you can reconsider your evaluation. If you have further questions, we are happy to address.

---

### Author Response · Authors · 2025-11-19
**Response to all reviewers**

Dear Area Chair, SPC, and Reviewers,

We are grateful for the reviewers' thoughtful comments and the time dedicated to evaluating our submission. Their insights have greatly strengthened our work. We have now revised the manuscript accordingly and submitted the updated PDF, along with this point-by-point summary of revisions. Key modifications within the manuscript are highlighted in blue for your convenience.

- Experiments: We have added additional experiments including: (1) combining DeFT with another quantization method, LoftQ (cf. Table 6); (2) evaluations on more challenging task MATH (cf. Table 8); (3) impact of different calibration data subsets on the final performance  (cf. Table 11).

- Presentation & Discussion: We have revised the notations and technical details (cf. Section 2), and added an algorithmic pseudocode for the truncation position search algorithm to improve the clarity of our method (cf. Appendix B). Besides, we have followed the reviewer’s suggestion to add more deeper discussion on the "compression-for-a-gain" phenomenon of our experimental results (cf. Section 3.1.1).

- Methodology: We have clarified the novelty, baseline comparison and fine-tuning efficiency of our method.

Sincerely,

Authors of submission #24949

---

### Comment · Area_Chair_VeRe · 2025-11-20
**Action Needed: Review Rebuttal and Update Evaluation**

Dear Reviewers,

Thank you, as always, for your valuable contributions and efforts. The authors have now submitted their rebuttal. Please take a moment to review it and provide any necessary follow-up actions, such as additional questions, clarification requests, or updates to your review.

Since the initial ratings ranged from 2 to 6, I kindly ask you to pay close attention to the perspectives of the other reviewers when preparing your final response.

Thank you again for your support.

---

### Meta-Review · Area_Chair_Bg9v · 2026-01-06

**Summary:**

This paper proposes a novel strategy for finetuning that reduces the number of model parameters updated during finetuning without affecting model quality.

**Reviewer Concerns:**

The main concerns raised by the reviewers were regarding the scope of the experimental evaluation (in terms of baselines and datasets), the significance of the amount of memory saved, and the clarity of presentation. The clarity of presentation was only raised by one reviewer, so I am discounting it. The significance of the memory saved was addressed by the response; it is higher than the reviewer's concern.

The biggest remaining concern is the scope of the experiments. While the response introduced some new experiments, they appear to be largely preliminary, and I do not believe they fully address the concerns that were raised. First, the only additional dataset that the authors include is the MATH dataset, which is qualitatively quite similar to GSM-8K (though it is harder). I believe that including a greater breadth of datasets would improve confidence in the approach, especially as a general fine-tuning methodology. More significantly, there were multiple reviewers who raised concerns about baselines. The reviewers only added a comparison to one additional PEFT, LoftQ; furthermore, I only saw results for accuracy, not for memory savings. I believe additional baselines along the lines suggested by reviewers (e.g., comparison to quantization) would make the paper significantly stronger.

**Reviewer Scores:**

Given that the concerns regarding the scope of the experiments were not fully addressed, and that these concerns were shared across multiple reviewers, I'm not sure the reviewers would have significantly changed their scores.

---

### Decision · Program_Chairs · 2026-01-26

Reject